# Genetic variation of *BnaA3.NIP5;1* expressing in the lateral root cap contributes to boron deficiency tolerance in *Brassica napus*

**Mingliang He[1,2], Sheliang Wang[2], Cheng Zhang[1,2], Liu Liu[3], Jinyao Zhang[4], Shou Qiu[1], Hong Wang[4], Guangsheng Yang[1], Shaowu Xue[3], Lei Shi[1,2], Fangsen Xu[1,2]***

**1** National Key Laboratory of Crop Genetic Improvement, Huazhong Agricultural University, Wuhan, China, **2** Microelement Research Centre, Huazhong Agricultural University, Wuhan, China, **3** College of Life Science and Technology, Huazhong Agricultural University, Wuhan, China, **4** Institute of Agricultural Resource and Regional Planning, CAAS, Beijing, China

* fangsenxu@mail.hzau.edu.cn

**Data Availability Statement:** All relevant data are within the manuscript and its Supporting Information files.

## Abstract

Boron (B) is essential for vascular plants. Rapeseed (*Brassica napus*) is the second leading crop source for vegetable oil worldwide, but its production is critically dependent on B supplies. *BnaA3.NIP5;1* was identified as a B-efficient candidate gene in *B. napus* in our previous QTL fine mapping. However, the molecular mechanism through which this gene improves low-B tolerance remains elusive. Here, we report genetic variation in *BnaA3. NIP5;1* gene, which encodes a boric acid channel, is a key determinant of low-B tolerance in *B. napus*. Transgenic lines with increased *BnaA3.NIP5;1* expression exhibited improved low-B tolerance in both the seedling and maturity stages. BnaA3.NIP5;1 is preferentially polar-localized in the distal plasma membrane of lateral root cap (LRC) cells and transports B into the root tips to promote root growth under B-deficiency conditions. Further analysis revealed that a CTTTC tandem repeat in the 5'UTR of *BnaA3.NIP5;1* altered the expression level of the gene, which is tightly associated with plant growth and seed yield. Field tests with natural populations and near-isogenic lines (NILs) confirmed that the varieties carried *BnaA3.NIP5;1^Q* allele significantly improved seed yield. Taken together, our results provide novel insights into the low-B tolerance of *B. napus*, and the elite allele of *BnaA3.NIP5;1* could serve as a direct target for breeding low-B-tolerant cultivars.

## Author summary

Boron (B) deficiency severely rapeseed (*Brassica napus*) yields in most high rainfall areas worldwide, and genetic improvement is an effective strategy for addressing the problem. Here we show that *BnaA3.NIP5;1*, encoding a boric acid channel, is a key determinant of the low-B tolerance in *B. napus*. Our results demonstrate that *BnaA3.NIP5;1* is preferentially located in the distal side plasma membrane of lateral root cap (LRC) cells and transports B into meristem zone to promote root growth under B limitation, which provide insights into the LRC's function in mineral nutrition. A CTTTC tandem repeat in the

**Funding:** This research was funded by the National Key Research and Development Program of China: 2016YFD0100700, National Natural Science Foundation of China (NSFC): 31772380, 31572185 and Fundamental Research Funds for the Central Universities of China: 2662019PY058, 2662019PY013 to F.S.X. and NSFC: 31670267, 31770283 to S.W.X. The funders had no role in study design, data collection and analysis, decision to publish, or preparation of the manuscript. The funders had no role in study design, data collection and analysis, decision to publish, or preparation of the manuscript.

**Competing interests:** The authors have declared that no competing interests exist.

5'UTR of *BnaA3.NIP5;1* altered the expression level of the gene, which is tightly associated with plant growth and seed yield under low-B conditions. The functional gene and elite allele could be useful in rapeseed breeding.

## Introduction

Boron (B) is an essential micronutrient for all higher plants. Plant roots take up B in the form of boric acid ($H_3BO_3$). However, the natural abundance of B is relatively low in the soil [1], and B is very leachable, especially in areas of high rainfall (South-East Asia, Brazil, China). The leaching of B from soil leads to a decrease in the availability of B to plants. B deficiency is a worldwide agricultural production problem that has been reported in the field for at least 132 crops from 80 countries [2]. B acts as a cross-link between the pectin polysaccharide rhamno-galacturonan II (RG-II) via borate-diol ester bonds in vascular plants and is necessary for plant growth [3–6]. B deficiency during early vegetative stages leads to slow growth and low biomass, whereas B deficiency can considerably diminish productivity [7]. Application of B fertilizer can alleviate B-deficiency problems, but borate rock is a non-renewable mineral resource. Thus, genetic improvement of B efficiency of crops is a promising and cost-efficient strategy in B-deficient regions.

Under low-B conditions, B can be taken up via two different mechanisms in plants. Channel-mediated facilitated transport via *NOD26-LIKE MAJOR INTRINSIC PROTEIN5;1* (*NIP5;1*) encodes a channel protein belonging to the aquaporin family [8]. Energy-dependent active transport against concentration gradients via AtBOR1, an anion co-exchanger that functions as an efflux transporter for xylem loading of B under B-limiting conditions [9]. Likely, the coordinated functions of both the channel protein and the transporter are essential for the growth of plants under B-limiting conditions [10]. Overexpression of *AtBOR1* or increased expression of *AtNIP5;1* can improve B-deficiency symptoms in *Arabidopsis* [11,12], but techniques for improving crop tolerance to low-B conditions have rarely been reported.

Rice *OsNIP3;1* is a homolog of *AtNIP5;1*, and the *Osnip3;1* mutant shows typical B-deficiency symptoms under B deficiency [13–15]. *OsBOR1* was reported to be involved in both B uptake and xylem loading [16]. In maize, *TLS1* encodes a protein that is a member of the aquaporin family and is co-orthologous to *AtNIP5;1*, and *RTE* encodes a B-efflux transporter that is co-orthologous to *AtBOR1*. Both *tls1* and *rte* mutants showed vegetative and reproductive defects in low-B soils [17,18]. It was recently reported that *BnaC4.BOR1;1c* is essential for the inflorescence development of rapeseed under B deficiency [19].

Allotetraploid rapeseed (*Brassica napus* L., AnAnCnCn, 2n = 38), one of the main oil crop species worldwide, has a high B demand and is highly sensitive to B deficiency [20,21]. Under B deficiency, *B. napus* exhibit evident growth defects in both vegetative and reproductive organs, including inhibited root growth, curved leaves, multiple branches, necrosis and protruding stigmas, all of which lead to severe losses in seed yield [22]. Therefore, one potential solution is the identification of target genes for low-B tolerance for breeding B-efficient rapeseed.

Previous studies have indicated that natural rapeseed varieties significantly vary in their tolerance to low-B conditions [23,24]. Quantitative trait locus (QTL) analysis revealed *BnaA3.NIP5;1* to be the candidate gene for *qBEC-A3a*, a major quantitative trait locus for low-B tolerance from B-efficient (low-B tolerant) Qingyou 10 (QY10) no difference was found in the amino acid sequence of *BnaA3.NIP5;1* between QY10 and the B-inefficient (low-B sensitive) Westar 10 (W10), while the expression level of *BnaA3.NIP5;1* was significantly higher in QY10

than in W10 [25,26]. However, the molecular mechanism of *BnaA3.NIP5;1* in response to B deficiency is still unknown. Here, we found that BnaA3.NIP5;1 is a boric acid channel responsible for B uptake into the root tips due to the specific distal polar localization in lateral root cap (LRC) cells. Further investigation revealed that genetic variation in the 5'UTR of *BnaA3.NIP5;1* dictates the distinct transcript abundance, thus leading to different B efficiency-dependent root growth and development and seed yields. Our study provides new insight into the biological role of *BnaA3.NIP5;1* for low-B tolerance and a novel perspective on the genetic improvement of B efficiency in *B. napus*.

## Results

### Differential expression of *BnaA3.NIP5;1* is responsible for low-B tolerance

To investigate the relationship between *BnaA3.NIP5;1* expression levels and low-B tolerance, we generated transgenic rapeseed expressing *BnaA3.NIP5;1*-GFP under control of the 2927 bp QY10 promoter (pQ::BnaA3.NIP5;1-GFP) in W10. Three independent pQ::BnaA3.NIP5;1-GFP transgenic lines were analyzed in the $T_2$ generation. qRT-PCR analysis revealed that the expression of *BnaA3.NIP5;1* in the roots of the three lines was significantly increased compared with that of the wild-type (WT) W10 (Fig 1B). Consequently, compared with W10, the transgenic lines presented longer roots and greater biomass under low-B (0.25 μM) conditions (Fig 1A and 1C–1E). In comparison, no significant difference was observed among these plants under normal-B (100 μM) conditions (S1A–S1C Fig). We further employed RNA interference (RNAi) to suppress the expression of *BnaA3.NIP5;1* in both QY10 (Q^RNAi) and W10 (W^RNAi). With decreased *BnaA3.NIP5;1* expression level (Fig 1G), the Q^RNAi transgenic plants exhibited severe growth defects compared with those of the QY10 plants under 0.25 μM B conditions (Fig 1F), with markedly reduced primary root length and biomass (Fig 1H–1J). No

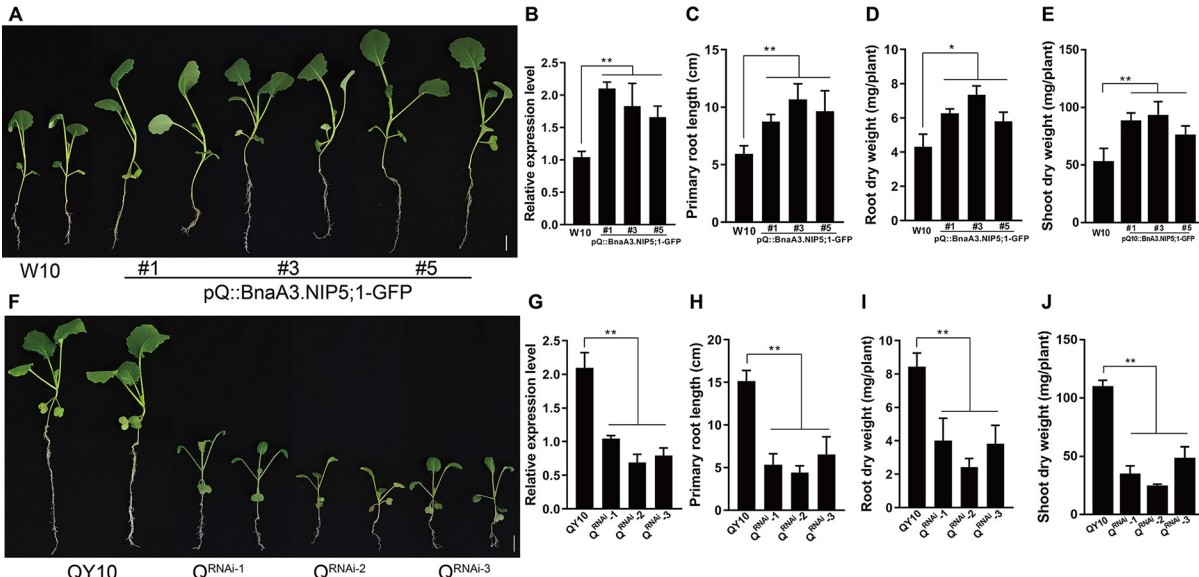

**Fig 1. Response of WT and transgenic plants to B deficiency.** (*A*) Phenotype of 15-d-old WT (W10) and pQ::BnaA3.NIP5;1-GFP transgenic plants grown under low-B (0.25 μM) conditions. Scale bar, 2 cm. (*B-E*) Relative expression of *BnaA3.NIP5;1* (*B*), primary root length (*C*), root dry weight (*D*) and shoot dry weight (*E*) of W10 and pQ::BnaA3.NIP5;1-GFP transgenic plants under low-B (0.25 μM) conditions. (*F*) Phenotype of 15-d-old WT (QY10) and QY10-RNAi lines grown under low-B (0.25 μM) conditions. Scale bar, 2 cm. (*G-J*) Relative expression of *BnaA3.NIP5;1* (*G*), primary root length (*H*), root dry weight (*I*) and shoot dry weight (*J*) of QY10 and QY10-RNAi lines under low-B (0.25 μM) conditions. The data presented are mean values with s.d. (three replicates in (*B* and *G*); six replicates in (*C, D, E, H, I* and *H*). * $P<0.05$, ** $P<0.01$ (Student's *t*-test).

noticeable difference was found under 100 μM B conditions (S1D–S1F Fig). We also noticed the expression of *BnaA3.NIP5;1* in the W[RNAi] transgenic plants was similar to that in W10, and both the W[RNAi] transgenic plants and W10 showed typical B-deficiency symptoms (S1G–S1J Fig), which is probably due to the low expression level of *BnaA3.NIP5;1* in W10.

To further investigate the effects of *BnaA3.NIP5;1* expression on seed productivity in *B. napus*, we compared the seed yield of Q[RNAi] transgenic plants and QY10 growing in pots at the maturity stage. The results showed that the per-plant seed yield of the Q[RNAi] transgenic plants was significantly lower than that of QY10 under low-B conditions (S2A Fig). However, there was no difference under high-B conditions (S2B Fig). Taken together, these data suggest increased expression of *BnaA3.NIP5;1* contributes to low-B tolerance in rapeseed during both the seedling and maturity stages.

## BnaA3.NIP5;1 functions as a boric acid channel and is expressed preferentially in LRCs

*BnaA3.NIP5;1* encodes a major intrinsic protein belonging to the aquaporin protein family [27,28]. Phylogenetic analysis revealed that BnaA3.NIP5;1 is a homolog of AtNIP5;1, with 92% amino acid identity (S3A and S3B Fig). To verify that BnaA3.NIP5;1 protein function in transporting $H_3BO_3$ across membranes, we expressed BnaA3.NIP5;1-GFP in *Xenopus* oocytes. The results showed that BnaA3.NIP5;1-GFP was localized on the plasma membrane of the oocytes (Fig 2A) and that oocytes carrying BnaA3.NIP5;1-GFP had a $^{10}$B concentration 3-fold higher than that of $H_2O$-injected oocytes (Fig 2B). These findings demonstrate that BnaA3.NIP5;1 has $H_3BO_3$ transport activity.

To investigate the expression pattern of *BnaA3.NIP5;1*, we generated pQ::BnaA3.NIP5;1-β-glucuronidase (GUS) transgenic plants in the W10 background. GUS activity was distinctly detected in the root tips (Fig 2E and 2F); specifically, cross-sections of the root tips revealed high GUS activity in the peripheral tissues (Fig 2G). The subcellular localization of BnaA3-NIP5;1 was examined using pQ::BnaA3.NIP5;1-GFP transgenic plants. Confocal imaging showed that BnaA3.NIP5;1-GFP was polarly located in the distal plasma membrane of LRC cells (Fig 2H–2J). Such an expression pattern was confirmed by an *in situ* RT-PCR assay (Fig 2C and 2D), as well as in *Arabidopsis* (S4A–S4C Fig). Collectively, these findings provide strong evidence that BnaA3.NIP5;1 was distinctly polar-localized in the distal plasma membrane of LRC cells and functions directly in $H_3BO_3$ uptake.

## BnaA3.NIP5;1 transports B into the root tips to promote root growth

To investigate the role of *BnaA3.NIP5;1* in *B. napus*, we performed short-term $^{10}$B accumulation assays of QY10, W10 and NIL$^Q$-W10 (NIL carrying the homozygous allele from QY10 in the W10 background). The results showed that the $^{10}$B concentration in both the shoots and roots of QY10 was lower than that in both the shoots and roots of W10 and NIL$^Q$-W10 (Fig 3A and 3B), while the $^{10}$B concentration was significantly increased in the root tips of QY10 and NIL$^Q$-W10 compared with those in W10 (Fig 3C). No significant difference was observed in $^{11}$B concentrations (S5A and S5B Fig). We further performed an *in situ* laser ablation-inductivity coupled plasma-mass spectrometry (LA-ICP-MS) assay to visualize the distribution of B in the root tips. Consistent with the results of the $^{10}$B accumulation assay in the root tip, the results showed that QY10 and NIL$^Q$-W10 had higher B concentrations in the root tips than did W10 at 0.1 μM B, especially in the cells in meristem region (Fig 3E); however, there was no distinct difference under 100 μM B (Fig 3F). On the basis of these results, we hypothesized that *BnaA3.NIP5;1* might facilitate root growth in *B. napus*. As expected, QY10 and NIL$^Q$-W10

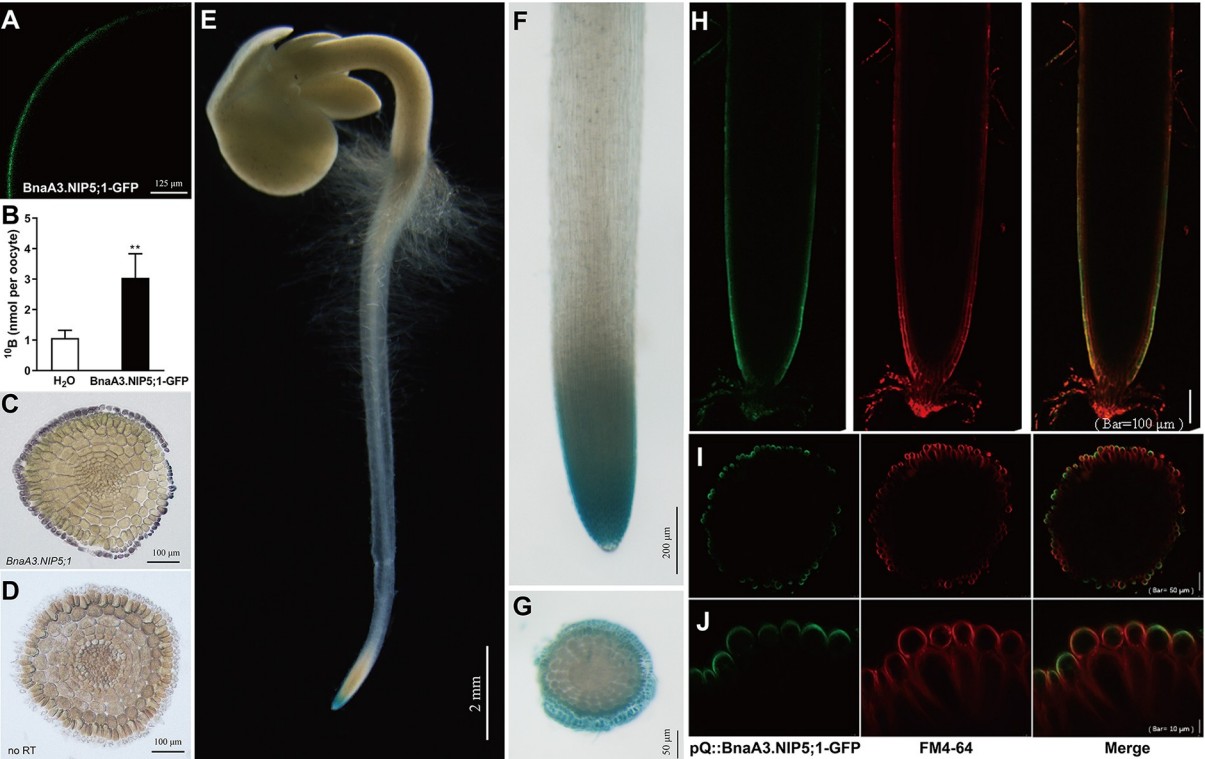

**Fig 2. Functional characterization and tissue and subcellular localization of BnaA3.NIP5;1.** (*A*) Subcellular localization of BnaA3.NIP5;1-GFP within the oocyte membrane. (*B*) Boric acid uptake in *Xenopus* oocytes injected with BnaA3.NIP5;1-GFP composed of $^{10}$B. The data presented are mean values with s.d. of four replicates. ** $P<0.01$ (Student's *t*-test). (*C* and *D*) RNA *in situ* RT-PCR of the cell-specific expression pattern of *BnaA3.NIP5;1* in root tip cross-sections. The dark blue signal indicated the presence of *BnaA3.NIP5;1* mRNA; no reverse transcription (no RT) conditions were used as negative controls. Similar results were observed in each of three independent experiments. (*E-G*) GUS staining of roots and cross-sections of root tips of pQ::BnaA3.NIP5;1-GUS transgenic plants. Similar results were observed in each of the three independent transgenic lines. (*H-J*) Confocal images of pQ::BnaA3.NIP5;1-GFP transgenic plants with vertical sections and cross-sections of the root tips; FM4-64 was used as a membrane-selective tracer. Similar results were observed in each of the three independent transgenic lines. All the rapeseed plants analysed were grown in solid media consisting of 0.1 μM B. Scale bars as shown.

had longer primary roots than did W10 at 0.1 μM B (Fig 3D and 3G), while there was no difference under 100 μM B (Figs 3H and S5C).

To further validate the role of *BnaA3.NIP5;1* in root growth, we generated transgenic plants expressing pQ::BnaA3.NIP5;1-GFP or pW::BnaA3.NIP5;1-GFP in the background of Arabidopsis mutant *nip5;1–1*. The primary roots of all the transgenic plants were longer than those of the *nip5;1–1* plants and compared with the pW::BnaA3.NIP5;1-GFP#n transgenic plants, the pQ::BnaA3.NIP5;1-GFP#n lines had higher *BnaA3.NIP5;1* expression levels and longer primary roots (Fig 4A–4C). By contrast, there was no significant difference in shoot growth among all these genotypes (Fig 4D). These results demonstrate that BnaA3.NIP5;1 transports B into root tips and specifically promotes root growth and development rather than facilitates B uptake in root vasculature tissues.

## A CTTTC tandem repeat in the 5'UTR alters the expression of *BnaA3. NIP5;1*

An approximately 3 kb promoter fragment containing the 5'UTR of *BnaA3.NIP5;1* was compared between QY10 and W10. Fifty-one single-nucleotide polymorphisms (SNPs), 4 InDels and one transposable element (TE) (differed in length between the two) were found between

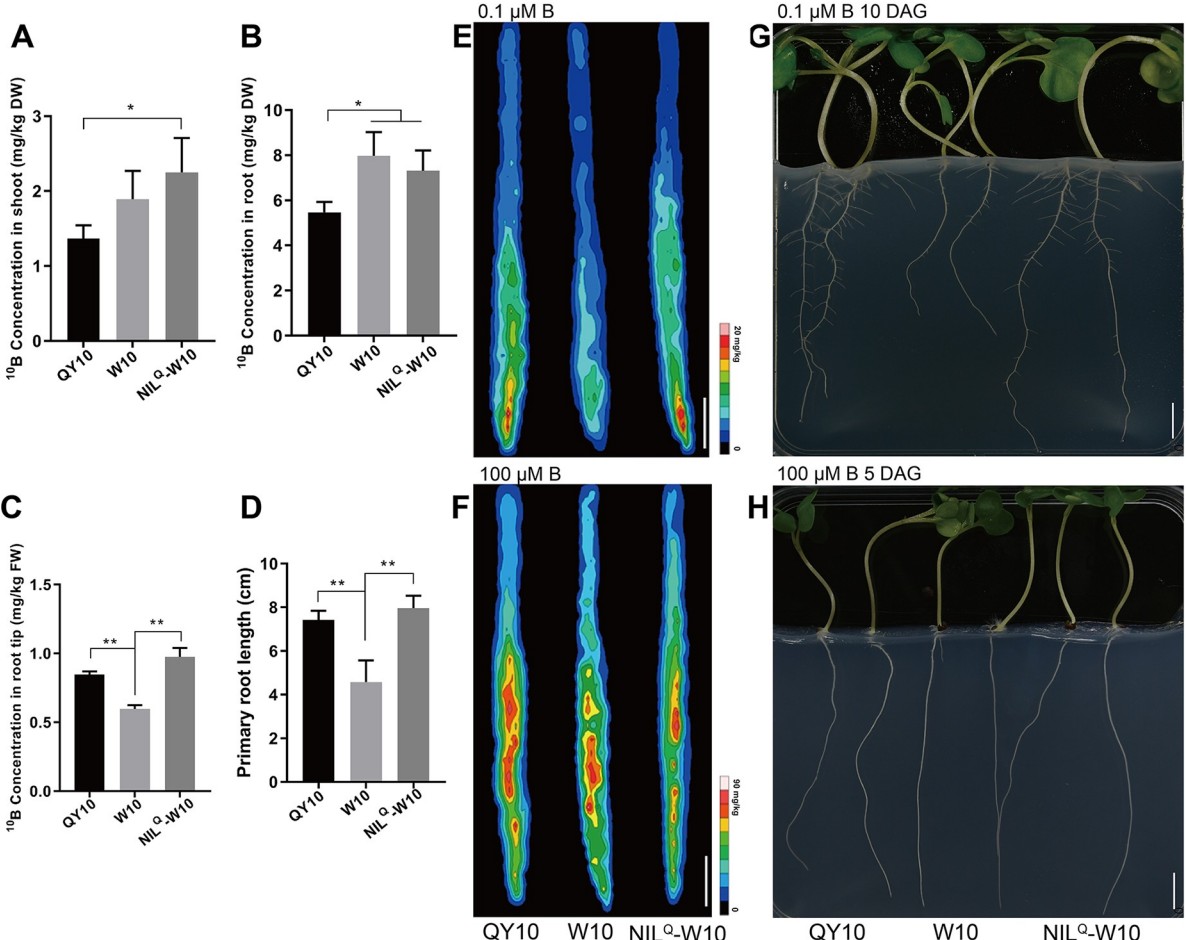

**Fig 3. BnaA3.NIP5;1 transports B into the root tips to promote root growth.** (*A-C*) $^{10}$B concentrations in the shoots (*A*), roots (*B*) and root tips (*C*) of QY10, W10 and NIL$^{Q}$-W10. In (*A* and *B*), QY10, W10 and NIL$^{Q}$-W10 seedlings were pre-cultured with 25 μM $^{11}$B for 15 d and then exposed to a solution without B for 1 d. The seedlings were subsequently exposed to 10 μM $^{10}$B for 1 h. In (*C*), the seedlings were pre-cultured in solid media without B for 5 d, after which the root tips were covered by a piece of solid media consisting of 10 μM $^{10}$B for 1 h, with 50 to 60 root tips constituting one replicate. After HNO$_3$ digestion, $^{10}$B and $^{11}$B were determined by ICP-MS. (*D*) Primary root length of QY10, W10 and NIL$^{Q}$-W10 seedlings grown on solid media consisting of 0.1 μM B for 10 d. (*E* and *F*) $^{10}$B distribution in the root tips of 5-d-old QY10, W10 and NIL$^{Q}$-W10 under 0.1 μM (*E*) and 100 μM (*F*) B according to LA-ICP-MS. Roots of QY10, W10 and NILQ-W10 seedlings grown on solid media consisting of 0.1 μM $^{10}$B and 100 μM $^{10}$B conditions for 5 d. The root tips were put into the laser ablation chamber and scanned together with the standard reference materials; $^{10}$B was determined by ICP-MS, and similar results were observed in each of three independent experiments. Scale bars, 100 μm. (*G* and *H*) Morphology of QY10, W10 and NIL$^{Q}$-W10 root growth. The rapeseed seedlings were grown on solid media consisting of 0.1 μM B for 10 d (*G*) and 100 μM B for 5 d (*H*). Scale bars, 1 cm. The data presented are mean values with s.d. Three replicates in (*A-C*); sixteen replicates in (*D*). * *P*<0.05, ** *P*<0.01 (Student's *t*-test).

the two promoter sequences (Figs 5A and 6). To identify the functional allelic variations in the promoter sequence, we generated a series of GUS constructs whose promoter sequence was truncated and transformed them into Columbia-0 (Col-0) plants (Fig 5B). GUS activity was subsequently detected in the tips of both primary roots and lateral roots of G1 (pQ2727::GUS) and G2 (pW3175::GUS) transgenic plants (S7 Fig), consistent with the expression pattern of *BnaA3.NIP5;1* in rapeseed (Fig 2E), and increased GUS activity was observed in the G1 transgenic plants compared with the G2 transgenic plants (Figs 5C and S7). For each construct, the GUS activity was measured in the roots of three independent lines. GUS activity in the G1 transgenic plants was significantly higher than that in the G2 transgenic plants, and transgenic lines carrying G3 (pQ2468::GUS) and G5 (pQ849::GUS) had significantly higher GUS activity

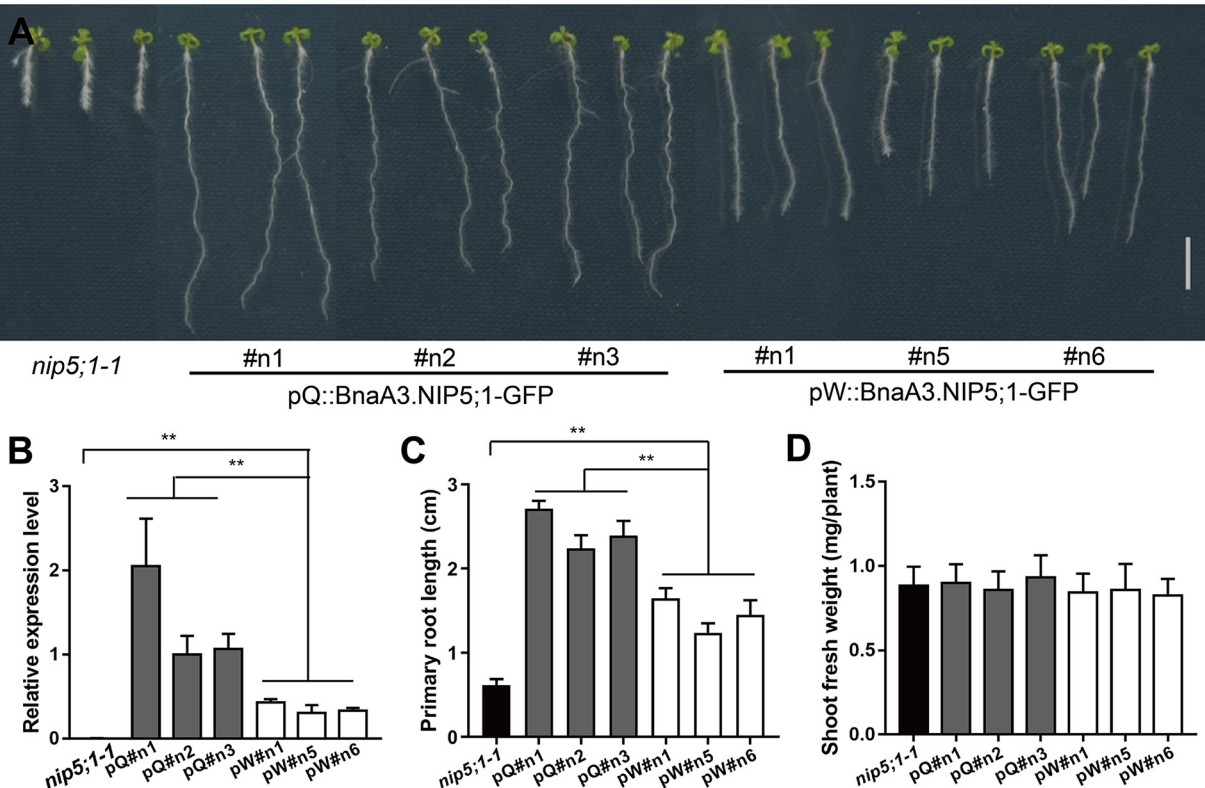

**Fig 4. *BnaA3.NIP5;1* promotes *Arabidopsis* root growth.** (*A*) Phenotype of 10-d-old *nip5;1–1*, pQ::BnaA3.NIP5;1-GFP#n (pQ#n) and pW::BnaA3.NIP5;1-GFP#n (pW#n) transgenic plants under 0.3 μM B conditions. Scale bar, 1 cm. (*B-D*) Relative expression level of *BnaA3.NIP5;1* (*B*), primary root length (*C*) and shoot fresh weight (*D*) of *nip5;1–1*, pQ::BnaA3.NIP5;1-GFP#n and pW::BnaA3.NIP5;1-GFP#n transgenic plants grown under 0.3 μM B conditions. The data presented are mean values with s.d. Three replicates in (*B*), twelve replicates in (*C*), and six replicates in (*D*). ** *P*<0.01 (Student's *t*-test).

compared with that of G4 (pW2706::GUS) and G6 (pW852::GUS) (Fig 5C and 5D). GUS activity in the G3 and G5 transgenic plants was similar to that in the transgenic lines carrying G1, while the GUS activity in G4 and G6 was similar to that in the G2 transgenic plants (Fig 5C and 5D). These results revealed that the polymorphisms from -849 to -1 were responsible for the different expression of *BnaA3.NIP5;1* between QY10 and W10.

Compared to W10, ten SNPs and 1 InDel (one CTTTC copy in QY10 and two tandem CTTTCs in W10) in the -464 to -1 5'UTR and 5 SNPs and 1 InDel in the -849 to -465 region were detected in the promoter region of QY10 compared with W10 (Fig 5E). We first verified whether the -849 to -1 sequence of *BnaA3.NIP5;1*[Q] (G5) could transcribe GUS in tobacco, which presented higher GUS activity than did the -852 to -1 sequence of *BnaA3.NIP5;1*[W] (G6) (Fig 5F and 5G). We exchanged the -465 to -849 promoter region between QY10 and W10 to generate G7 and G8 constructs (Fig 5F). Still, no effects on GUS activity were observed (Fig 5G), suggesting that polymorphisms from -464 to -1 are responsible for the different expression level of *BnaA3.NIP5;1* between QY10 and W10. We then deleted one CTTTC copy from the 5'UTR[W] (G9) and inserted one CTTTC copy into the 5'UTR[Q] (G10) to generate G9 and G10 constructs, respectively, and compared the GUS activity of G9 and G10 (Fig 5F). The deletion of one CTTTC copy from the 5'UTR[W] (G9) significantly enhanced GUS activity, and the insertion of one copy of CTTTC into the 5'UTR[Q] (G10) largely decreased GUS activity (Fig 5G). Taken together, these data indicate that a tandem repeat of CTTTC in the 5'UTR is responsible for the different expression of *BnaA3.NIP5;1*.

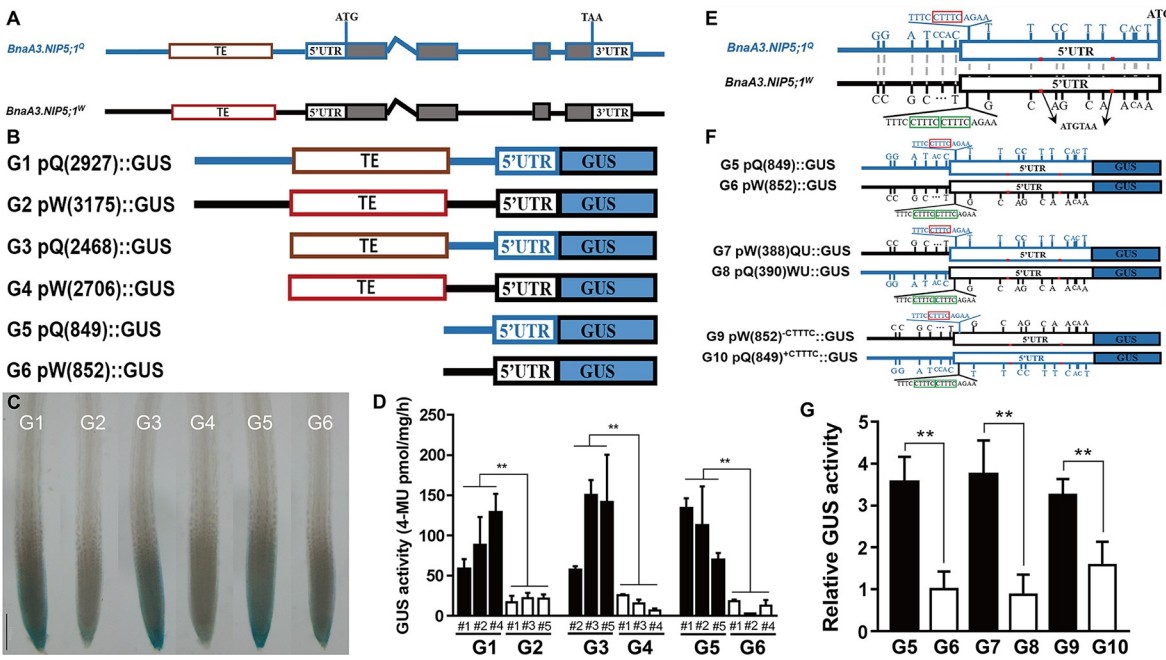

**Fig 5. A CTTTC tandem repeat in the 5'UTR confers the different expression of *BnaA3.NIP5;1*.** (*A*) Schematic diagram of *BnaA3. NIP5;1* in QY10 and W10. The white boxes represent 5'UTRs and 3'UTRs, grey boxes represent exons, the lines between grey boxes represents introns, and the brown and red boxes represent different TEs from QY10 and W10, respectively. (*B*) Constructs used for *Arabidopsis* transformation. G1 pQ(2927)::GUS contains the 2927 bp promoter of QY10. G2 pW(3175)::GUS contains the 3175 bp promoter of W10, and G3-G6 have a 5' series deletion as indicated. (*C* and *D*) GUS staining and activity of transgenic plants harbouring G1-G6 constructs. Seedlings of the G1-G6 transgenic plants were grown on solid media consisting of 0.3 μM B for 10 d. Scale bar, 100 μm. (*E*) Variations in the 849 bp fragment in the promoters of QY10 and W10. CTTTC tandem repeats are indicated by the red box in the *BnaA3.NIP5;1^Q* promoter and the green box in the *BnaA3.NIP5;1^W* promoter, respectively. The red bars indicate uORFs, and minus signs indicate nucleotide deletions and polymorphisms. (*F*) Constructs used for tobacco transformation. The 5'UTR and CTTTC repeat in G5/ G6 were exchanged to generate G7/G8 and mutated to generate G10/G9 constructs. (*G*) Relative GUS activity of different constructs (G5-G10), (CaMV) 35S-luciferase was co-transfected and used as an internal control. The data presented are mean values with s.d. Three replicates were used in (*D*), and six replicates were used in (*G*). ** $P < 0.01$ (Student's *t*-test).

## *BnaA3.NIP5;1^Q* allele improves seed yield under low-B conditions

To verify the contribution of the *BnaA3.NIP5;1^Q* allele to seed yield in rapeseed, we further introgressed the tolerant allele of *BnaA3.NIP5;1* from QY10 into the cytoplasmic male-sterile (CMS) restorer line L-135R through five generations backcrossing with L-135R. After two additional generations of self-pollination, we ultimately obtained BC₅F₃ plants (NIL^Q-L135R) carrying homozygous *BnaA3.NIP5;1^Q* alleles. Furthermore, W10, NIL^Q-W10, L-135R and NIL^Q-L135R were used to evaluate the low-B tolerance in field trials with and without B fertilizer. W10 and L-135R showed severe reproductive development defects with no B supply, reflected by fewer and seedless siliques ([Fig 6B](), [6D and 6E]()), while no B-deficiency symptoms were observed for NIL^Q-W10 and NIL^Q-L135R ([Fig 6A and 6B]()). The per-plant seed yield of NIL^Q-W10 and NIL^Q-L135R was significantly higher than that of W10 and L-135R, respectively, in the no B supply treatment ([Fig 6C and 6F]()). Taken together, these results indicate the selection of *BnaA3.NIP5;1^Q* can effectively improve seed yield under low-B conditions.

## *BnaA3.NIP5;1^Q* haplotype confers high *BnaA3.NIP5;1* expression and low-B tolerance to the natural population

To investigate the association between the variation in *BnaA3.NIP5;1* and B-deficiency tolerance, 10 B-efficient and 19 B-inefficient rapeseed varieties were selected from a natural

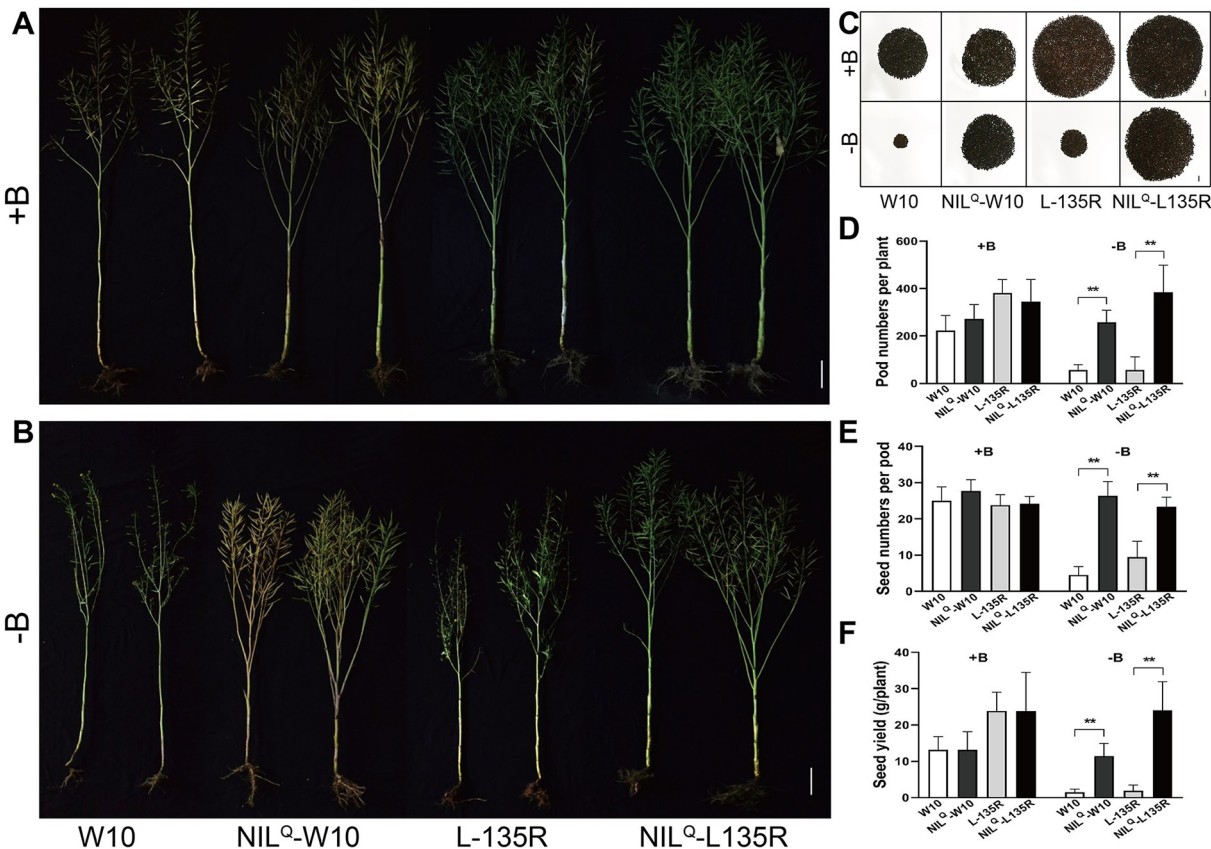

**Fig 6. The *BnaA3.NIP3;1^Q* allele improves low-B tolerance and seed yields of *B. napus*.** (*A* and *B*) Phenotypes of W10, NIL^Q-W10, L-135R and NIL^Q-L135R with B fertilization (+B) (*A*) and without B fertilization (-B) (*B*). Scale bar, 20 cm. (*C*) Total seeds per plant of W10, NIL^Q-W10, L-135R and NIL^Q-L135R with B fertilization (+B) and without B fertilization (-B). Scale bar, 1 cm. (*D-F*) Statistical comparisons of pod number per plant (*D*), seed number per pod (*E*), and seed yield per plant (*F*). The data presented are mean values with s.d. Twelve biological replicates in (*D-F*). ** *P*<0.01 (Student's *t*-test).

population comprising 210 rapeseed accessions [29]. We re-sequenced the -849 to -1 fragment of the *BnaA3.NIP5;1* promoter from the 29 varieties together with QY10, W10 and NIL^Q-W10. Based on the CTTTC tandem repeat, these 32 lines were classified into 2 groups, 13 of which presented the QY10 haplotype, with a single CTTTC copy (*BnaA3.NIP5;1^Q*), and 19 of which showed the W10 haplotype, with two tandem CTTTCs (*BnaA3.NIP5;1^W*) (S8 Fig). The expression levels of *BnaA3.NIP5;1* in these plants were subsequently examined and compared with the *BnaA3.NIP5;1^W* haplotype, the *BnaA3.NIP5;1^Q* haplotype presented increased expression under low-B conditions (Fig 7A and 7B). In agreement with these results, the primary roots of these varieties carrying *BnaA3.NIP5;1^Q* were longer than those of the varieties presented the *BnaA3.NIP5;1^W* haplotype (Figs 7C, 7D and S9A). Furthermore, the varieties presented *BnaA3.NIP5;1^W* haplotype showed severe reproductive development defects, and the per-plant seed yield of the varieties presenting the *BnaA3.NIP5;1^Q* haplotype were significantly higher than those presenting the *BnaA3.NIP5;1^W* haplotype (Figs 7F and 7G and S9B). The increased *BnaA3.NIP5;1* expression levels in the representative varieties were positively correlated with primary root length and seed yield under low-B conditions (Fig 7E and 7H). Overall, the genetic variations of *BnaA3.NIP5;1* in the different rapeseed varieties further confirmed that the CTTTC tandem repeats within the *BnaA3.NIP5;1* 5'UTR directly regulates its expression, ultimately affecting primary root length and productivity under B-deficiency conditions.

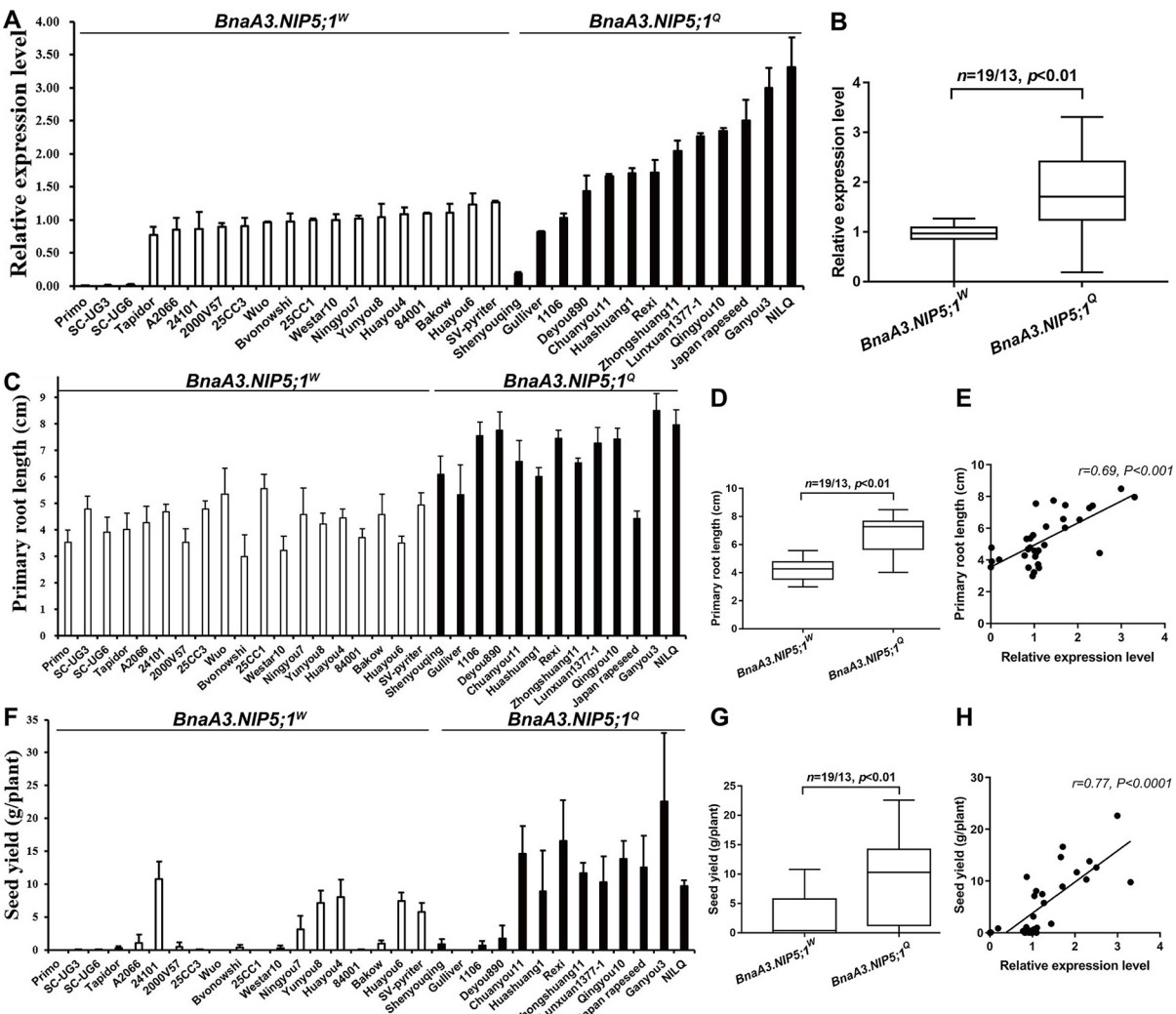

**Fig 7. A tandem CTTTC repeat modulates *BnaA3.NIP5;1* expression and low-B tolerance.** (*A*) Analysis of *BnaA3.NIP5;1* expression in the roots of representative varieties. Seedlings of the representative varieties were grown on solid media consisting 0.1 B for 10 d; afterward, the roots were sampled, and the expression level was determined by qRT-PCR. (*B*) The expression levels of *BnaA3.NIP5;1* of the different varieties were grouped according to the *BnaA3.NIP5;1^Q* and *BnaA3.NIP5;1^W* haplotypes. (*C*) Primary root length of representative varieties. Seedlings of the representative varieties were grown on solid media consisting of 0.1 B for 10 d. (*D*) The primary root lengths of different varieties were grouped according to the *BnaA3.NIP5;1^Q* and *BnaA3.NIP5;1^W* haplotypes. (*E*) Correlations between primary root length and *BnaA3.NIP5;1* expression level among different varieties under low-B conditions. (*F*) Per-plant seed yield of representative varieties grown in a B-deficient plot in the field. (*G*) Per-plant seed yield of representative varieties with the *BnaA3.NIP5;1^W* or *BnaA3.NIP5;1^Q* haplotype. (*H*) Correlations between per-plant seed yield and *BnaA3.NIP5;1* expression level among the different varieties under low-B conditions. In (*B*, *D* and *G*), the data represented are mean values of three replicates with s.d., the box shows the median and the lower and upper quartiles. The numbers (*n*) of each haplotype and associated *P* value (Student's *t*-test) are shown above the graph. In (*E* and *H*), the *r* and *p* values were determined by Pearson correlation analysis. Six replicates were used for the primary root length data, and twelve replicates were used for the seed yield data.

## Discussion

Crop production depends on nutrient uptake, and the reduction in crop yields caused by nutrient deficiency is an important agronomical problem worldwide. A previous study indicated that genetic variations in low-B tolerance exist among different rapeseed varieties [23,24], providing opportunities to improve low-B tolerance in B-inefficient rapeseed by introducing the relevant gene(s) from B-efficient varieties. In this study, we identified the B efficiency-related gene *BnaA3.NIP5;1*, which is expressed in LRC cells, facilitates B uptake into

root tips (Figs 2H and 3E). A CTTTC copy deletion within the 5'UTR increased *BnaA3.NIP5;1* expression levels subsequently promoted root growth and increased seed yields under B limitation (Fig 7A, 7C and 7F). Notably, the *BnaA3.NIP5;1^Q* allele effectively improves seed yields under low-B conditions (Fig 6F), indicating that *BnaA3.NIP5;1* could serve as a novel target in rapeseed breeding for improved low-B tolerance.

Normal plant growth requires an adequate B supply. Under B deficiency, the first symptoms of B deficiency occur in the growing tips of plants; these symptoms include root growth inhibition and shoot apical necrosis [21,30–32]. Thus, sufficient amounts of B must be available for developing tissue. A mathematical model was developed that predicted that the QC region has a high B concentration [33]. However, the mechanisms by which B is taken up or transported in root and shoot apices are still unclear. In the present study, our results showed that *BnaA3.NIP5;1* is expressed specifically and polarly localized in the distal plasma membrane of LRC cells to promote B uptake into root tips (Fig 2H), which is necessary for root growth. Under low-B conditions, QY10 with high *BnaA3.NIP5;1* expression levels in the LRC cells accumulated more B in the meristem region, which led to better root growth and development than those of W10 under B-deficiency conditions (Fig 3E and 3G). Furthermore, the results of the transgenic plants *nip5;1–1* expressing *BnaA3.NIP5;1* indicated that B taken up by BnaA3.NIP5;1 was used only for root growth and was not translocated to the shoots (Fig 4A). However, transgenic *nip5;1–1* still had a growth defect, which indicates *NIP5;1* expressing in the differential region is also important for plant normal growth under B limitation. It was reported that *AtBOR2* also had a higher expression level in the LRC cells and with a proximal polar localization [10]. This suggests that *BnaA3.NIP5;1* may cooperate with other B transporters to maintain high B concentration in the root meristem region under B-limitation. We conclude that B taken up by LRC cells is used only for local root growth and that high B concentrations are maintained in the meristem region, which is important for root development and growth under B deficiency. With the well-developed root, other boron related transporters uptake more B into roots and jointly promote shoot growth with *BnaA3.NIP5;1*.

Growing roots vary both anatomically and physiologically along their longitudinal axes [34]. Nutrient uptake also varies across the different developmental zones. Generally, the rate of ion uptake per unit root length decreases with increasing distance from the root apex) 34]. It was recently reported that LRC cells are important for Pi uptake, but they have not been shown to influence root growth [35]. The results from our study clearly show that B uptake into root tips is essential for root growth under low-B conditions (Fig 3E and 3G). Previous studies have also demonstrated the presence of transporters of nitrate (NRT1.1) [36], potassium (ATKT3) [37] and iron (IRT1) [38] ions in the root tips. However, the role of these proteins in nutrition is remained to be elucidated.

In *Arabidopsis*, AtNIP5;1 has been shown to be localized in the distal plasma membrane of both LRC cells and epidermal cells of roots [8,39]. A ThrProGly (TPG) repeat in the N-terminus of AtNIP5;1 is essential for AtNIP5;1 polar localization [40]. Two TPG repeats were found in the N-terminus of BnaA3.NIP5;1 (S3B Fig), and our results show that BnaA3.NIP5;1 localized in the plasma membrane in a polar manner, like AtNIP5;1 is. The upstream open reading frames (uORFs) within the 5'UTR of *AtNIP5;1* induce B-dependent *AtNIP5;1* mRNA degradation is important for plant growth under high-B conditions [41,42]. Two conserved uORFs were found in the 5'UTR of *BnaA3.NIP5;1* (Figs 5F and S10), which indicates that *BnaA3. NIP5;1* mRNA degradation under high-B conditions is similar to *AtNIP5;1* mRNA degradation. Transposons have been reported to produce a wide variety of changes in plant gene expression [43]. However, we found that the different TE insertion sequences do not alter the gene expression levels or the expression patterns in QY10 and W10. The 849 bp promoter upstream of ATG is responsible for gene expression, and a CTTTC deletion increased *BnaA3.*

*NIP5;1* expression levels, which improved low-B tolerance in QY10. No CTTTC element was found within the promoter of *AtNIP5;1* (S10 Fig), which suggests that CTTTC is unique to *BnaA3.NIP5;1* regulation in *B. napus*. However, the activities of pQ::BnaA3.NIP5;1-GFP and pW::BnaA3.NIP5;1-GFP differed when they were expressed in nip5;1–1 (Fig 4), it is possible that *Brassica napus* and *Arabidopsis* show the same trans-factor or transcript factor to regulate the CTTTC element, and further study can be carried out in *Arabidopsis* to found the upstream regelation element of *BnaA3.NIP5;1*.

Overall, *BnaA3.NIP5;1* and its elite allele can serve as direct targets for genetic improvement of low-B tolerance in rapeseed breeding. *BnaA3.NIP5;1* expressing in LRC cells for B uptake into root tips affords greatly increased seed yields under B deficit, highlighting the importance of nutrient uptake in root tips. The results in this study may provide a new way for improving rapeseed low-B tolerance and improving the efficiency of other nutrients.

## Methods

### Plant materials and growth conditions

The following plants were used in this study: the rapeseed (*B. napus*) cultivars W10 and QY10 and a CMS restorer line, L-135R; *Arabidopsis* (*Arabidopsis thaliana*) Col-0; and the *Arabidopsis nip5;1–1* mutant.

For long-term hydroponic cultivation, seeds of rapeseed were germinated on a piece of moist gauze submerged in ultrapure water (18.25 MΩ·cm) in a black plastic tray. After they germinated, uniform seedlings were transplanted into 10 L black plastic containers filled with Hoagland's solution [44] consisting of 5 mM $KNO_3$, 5 mM $Ca(NO_3)_2$, 2 mM $MgSO_4$, 1 mM $KHPO_4$, 50 μM FeEDTA, 9 μM $MnCl_2$, 0.8 μM $CuSO_4$, 0.8 μM $ZnSO_4$ and 0.1 μM $Na_2MO_4$. $H_3BO_3$ (100 μM) was used for normal-B treatment, and $H_3BO_3$ (0.25 μM) was used for low-B treatment. The nutrient solutions were refreshed every 3 d. The seedlings were grown in a greenhouse under a 16 h light (24˚C)/8 h dark (22˚C) photoperiod with an approximately 300 μM $m^{-2}$ $s^{-1}$ photon density.

For plate culture, seeds of *B. napus* and *Arabidopsis* were surface-sterilized for 15 min with 1% NaClO (w/v), rinsed with ultrapure water (18.25 MΩ·cm), chilled at 4˚C for 2 d in the dark, and then sown onto solid media for plate culture. The plant growth medium was MGRL media [45], consisting of 1.75 mM sodium phosphate buffer (pH 5.8), 1.5 mM $MgSO_4$, 2 mM $Ca(NO_3)_2$, 3 mM $KNO_3$, 67 μM $Na_2EDTA$, 8.6 μM $FeSO_4$, 10.3 μM MnSO4, 1 μM $ZnSO_4$, 24 nM $(NH_4)_6Mo_7O_{24}$, 130 nM $CoCl_2$, 1 μM $CuSO_4$, 1% sucrose and 1% gellan gum. $H_3BO_3$ (100 μM) was used for normal-B treatment, $H_3BO_3$ (0.1 μM) was used for rapeseed low-B treatment, and $H_3BO_3$ (0.3 μM) was used for *Arabidopsis* low-B treatment. The seedlings were grown in a growth chamber at 22˚C under a 16 h light/8 h dark photoperiod.

### RNA extraction and qRT-PCR analyses

Total RNA was extracted using an RNA extraction kit (Promega). The concentration of RNA was subsequently determined by a NanoDrop 2000 (Thermo Fisher). cDNA was prepared using Rever Tra Ace qPCR RT Master Mix with gDNA Remover kit (Toyobo). Quantitative real-time PCR assays were performed on a Real-time PCR Detection System (Applied Biosystems) in a 384-well plate via SYBR Green PCR (Toyobo). The $2^{-\Delta\Delta ct}$ quantification method was used, and the variation in expression was estimated for three biological replicates.

### Transgenic rapeseed construction and phenotypic analyses

An approximately 3 kb upstream regulatory sequence was amplified from the genomic DNA template of *BnaA3.NIP5;1$^W$* and *BnaA3.NIP5;1$^Q$*, and the coding sequence of *BnaA3.NIP5;1*

was amplified from a complementary DNA (cDNA) template fused in frame with GFP and then cloned into a pBI121 binary vector to generate pQ::BnaA3.NIP5;1-GFP constructs. The resulting vector was introduced into the B-inefficient variety W10 via *Agrobacterium*-mediated hypocotyl transgenic transformation. Additionally, *BnaA3.NIP5;1*-specific 312 bp sense and antisense fragments were amplified from the cDNA template and cloned into a pFGC5941binary vector to generate an RNAi construct. The resulting vector was transformed into QY10 and W10, yielding Q[RNAi] and W[RNAi] transgenic seedlings. Positive transgenic seedlings were identified in each generation via insertion-specific PCR analyses. The sequences of the primers used for vector construction and transgenic plant identification are listed in S1 Table.

Independent homozygous transgenic $T_2$ lines were grown as described above. The roots of both transgenic and WT seedlings were subsequently washed. After measuring the primary root length, the samples were dried at 65˚C for 72 h to obtain the shoot and root dry weights. The *BnaA3.NIP5;1* expression level was determined via qRT-PCR from root samples obtained from hydroponically cultivated 15-d-old rapeseed seedlings and normalized to the rapeseed *EF1α* and *Tubulin* internal control gene expression levels.

## [10]B uptake in *Xenopus laevis* oocytes

The coding DNA sequence (CDS) of *BnaA3.NIP5;1-GFP* was amplified from a pQ::BnaA3. NIP5;1-GFP vector, cloned into a pT7Ts *X. laevis* oocyte expression vector between the restriction sites *BglII* and *SpeI* and then linearized with *BamHI*. Capped mRNA was synthesized in vitro using an mMESSAGE mMACHINE kit (Ambion, AM1340). *X. laevis* oocytes were injected with 46 ng of BnaA3.NIP5;1-GFP cRNA and then cultured in ND96 media for 2 d for GFP observations via confocal microscopy (TCS SP8, Leica). GFP signal-positive oocytes or water-injected negative controls were collected from six-well plates filled with 5 ml of ND96 media, and then the ND96 media was removed and replaced with B-ND96 media consisting of 5 mM [10]B. After a 30 min incubation at 18˚C, each sample was rinsed five times with ice-cold ND96, and 8–11 oocytes were collected in a 2 ml tube and frozen at -20˚C until sampled for elemental analysis. The oocytes were digested with $HNO_3$ at a maximum temperature of 110˚C in plastic tubes, and the [10]B concentration was analyzed via inductively coupled plasma-mass spectrometry (ICP-MS, 7700X; Agilent Technologies). Four replicates of oocytes were used for [10]B uptake assays. The sequences of the primers used are listed in S1 Table.

### *In situ* RT-PCR and GUS staining

Rapeseed root samples from 5-d-old seedlings grown under B-deficiency conditions were fixed with a solution consisting of 63% (v/v) ethanol, 5% (v/v) acetic acid and 2% (v/v) formaldehyde for 4 h, embedded into 5% (w/v) agarose and then sectioned to 50 μm. *BnaA3.NIP5;1 in situ* RT-PCR flowed method with the modifications of Athman et al. (2014) [46]. The samples were stained using BM purple AP substrate (Roche) for 30 min, washed in an orderly manner with washing buffer, mounted in 40% (v/v) glycerol and then observed under a microscope (Nikon DS-Ri 2).

The pQ::BnaA3.NIP5;1 fragment was amplified from the pQ::BnaA3.NIP5;1-GFP vector and cloned into pBI121 to generate pQ::BnaA3.NIP5;1-GUS constructs. The resulting vectors were transformed into W10 rapeseed. The pQ::BnaA3.NIP5;1-GUS transgenic seedlings were incubated in a solution of 1 mg ml$^{-1}$ 5-bromo-4-chloro-3-indolyl-β-D-glucuronic acid (X-gluc), 100 mM sodium phosphate (pH 7.0), 0.5 mM $K_3Fe(CN)_6$, 0.5 mM $K_4Fe(CN)_6$, 10 mM $Na_2EDTA$, 0.1% (v/v) Triton X-100 and 20% (v/v) methanol at 37˚C in the dark for 1 h. After incubation, the chlorophyll was removed using 75% ethanol, and images were taken using stereomicroscope (Olympus SZX18).

## Tissue, subcellular localization assay and fluorescence intensity measurements

The $T_2$ generation of pQ::BnaA3.NIP5;1-GFP transgenic plants was used for tissue and subcellular assays. The transgenic plants were grown on 0.1 μM B MGRL solid media for 4–6 d. N-(3-Triethylammoniumpropyl)-4-(6-(4-(diethylamino) phenyl) hexatrienyl) (FM4-64) was used as an endocytic tracer. Confocal imaging was performed via confocal microscopy (TCS SP8, Leica). The pQ::BnaA3.NIP5;1-GFP and pW::BnaA3.NIP5;1-GFP constructs were transformed into *Arabidopsis* ecotype Col-0 plants and *nip5;1–1* mutants via the floral-dip method used for tissue, subcellular localization assays and fluorescence intensity measurements. Transgenic *Arabidopsis* plants were grown on 0.3 μM B MGRL solid media for 4–6 d, and confocal imaging was performed as described above.

## $^{10}$B boric acid uptake activity in the roots and root tips

Fifteen-day-old QY10, W10 and NIL$^Q$-W10 seedlings pre-cultured with 25 μM $^{11}$B (Cambridge Isotope Laboratories) were exposed to a solution containing 0 B for 1 d. The seedlings were subsequently exposed to 10 μM $^{10}$B (Cambridge Isotope Laboratories) for 1 h, after which the roots were washed with ultrapure water three times. Shoot and root samples were collected, separated and then dried at 65°C for 72 h. The samples were then digested with $HNO_3$ at a maximum temperature of 110°C in plastic tubes and the resulting digestions were analyzed via ICP-MS (7700X; Agilent Technologies). To investigate B uptake in root tips, rapeseed seedlings were pre-cultured in 0 B MGRL solid media for 5 d. Afterward, 1×1 cm pieces of MGRL solid media consisting of 10 μM $^{10}$B were applied such that the root tips were covered for 1 h. The roots were then washed three times with ultrapure water, after which the root tips (5 mm) were excised by the use of a razor. The fresh weight was immediately recorded for 50 to 60 root tips, which constituted one replicate. After digestion, the $^{10}$B concentration was determined as described above.

## Phenotyping the primary root growth of rapeseed and *Arabidopsis* under B deficiency

Seeds of QY10, W10, NIL$^Q$-W10 and the representative varieties selected from the natural population were put atop solid media, and the seedlings were grown on vertically oriented solid media in a growth chamber. Under the 100 μM B conditions, 5 d was enough for the primary roots of the rapeseed seedlings to reach the bottom of the plastic dish, while under the 0.1 μM B conditions, the seedlings were allowed to grow for 10 d before images were collected and the primary root length was measured.

Seedlings of the *Arabidopsis* pQ::BnaA3.NIP5;1-GFP#n and pW::BnaA3.NIP5;1-GFP#n $T_3$ transgenic lines were grown on MGRL solid media consisting of 0.3 μM B. After 10 d of growth in the growth chamber, the primary root length was measured, and the expression of *BnaA3* was measured. *NIP5;1* expression in the roots was determined via qRT-PCR and normalized to the *Arabidopsis EF1α* and *Actin* internal control gene expression.

## Analysis of B distribution using laser ablation-inductively coupled plasma-mass spectrometry (LA-ICP-MS)

Seedlings of rapeseed were grown on B-deficient solid media for 10 d, after which the plant material was collected and ground into powder after drying. Afterward, 0, 20, 40, 100, 200, 400, or 1000 μl of 10 mg L$^{-1}$ $^{10}$B were added to the 200 mg powder, respectively, and were mixed together well. After absorption for 48 h, the mixture was re-dried and ground to a

powder. After digestion, the $^{10}$B concentration was determined as described above. Fifty milligrams of powder were then compressed under 8 atm to generate standard reference material (S11 Fig).

Seedlings of QY10, W10 and NIL$^Q$-W10 were grown in solid media under 0.1 μM $^{10}$B and 100 μM $^{10}$B conditions for 5 d. The roots were washed three times with ultrapure water, and the root tips (1.5 to 2 cm in length) were excised and then affixed to slides, which were then dried overnight at -20˚C. LA-ICP-MS analysis was subsequently performed with a laser ablation system (New Wave Research UP 213) equipped with a Nd: YAG laser (wavelength, 213 nm; repetition frequency, 20 Hz; spot size, 50 μm; scan speed, 20 μm s$^{-1}$; energy output: 50%; He carrier flow rate, 900 ml min$^{-1}$). The root tips were put into the laser ablation chamber and scanned together with the standards. Element image transformation was performed by Surfer 11 software. Three biological replications of each sample were analyzed, each of which showed similar results.

## GUS activity and transient gene expression assays

The *BnaA3.NIP5;1* promoter regions from W10 and QY10 were serially deleted and cloned into pBI121 vectors to generate G1-G6 constructs, and the resulting vectors were transformed into *Arabidopsis* ecotype Col-0 by the floral-dip method. Independent homozygous T$_3$ transgenic lines were grown in MGRL solid media consisting of 0.3 μM B for 10 d. Total protein extraction and quantitative GUS activity assays were conducted as described by Jefferson et al. (1987) [47]. The protein concentration was determined using a Bio-Rad protein assay kit (Bio-Rad), and the fluorescence intensity was measured on a Spark 20M multimode plate reader (Tecan). Addition mutations in G5 and G6 were investigated to generate G7-G10 constructs, and the resulting vectors were transformed into tobacco by *Agrobacterium*-mediated transformation for transient gene expression assays. After transfection, GUS activity was determined as described above. Luciferase was co-transfected and used as an internal control to normalize the data.

## NIL construction

The homozygous NIL$^Q$-W10 line was previously derived from a QY10 donor parent and a W10 recurrent parent (25, 26). With respect to NIL$^Q$-L135R, F$_1$ seeds were obtained by crossing L-135R with QY10. *BnaA3.NIP5;1* in each generation was genotyped, and the heterozygous plants were backcrossed with L-135R to the BC$_5$F$_1$ generation. The BC$_5$F$_1$ plants were then self-pollinated to obtain BC$_5$F$_2$ plants, which were self-pollinated to get BC$_5$F$_3$ plants for further analysis. Morphological differences and B-deficiency tolerance were compared between NILs homozygous for *BnaA3.NIP5;1*$^Q$ and recurrent parents for *BnaA3.NIP5;1*$^W$.

## Evaluation of B-deficiency tolerance under pot and field conditions

For pot cultivation, each pot contained 7 kg of grey purple sandy soil. The basic agrochemical characteristics of the soil were as follows: pH (1:1 soil: H$_2$O (w/v)), 7.7; organic matter, 1.33 g kg$^{-1}$; total nitrogen (N), 0.25 g kg$^{-1}$; total phosphorus (P), 72 mg kg$^{-1}$; and hot water-soluble B, 0.10 mg kg$^{-1}$. Two B treatments, 1 mg B kg$^{-1}$ soil (HB) and 0.25 mg B kg$^{-1}$ soil (LB), were applied, with four replicates per treatment. The plants were irrigated with ultrapure water.

For field trials, the B-deficiency tolerance of QY10, W10, NIL$^Q$-W10, NIL$^Q$-L135R and the representative varieties were compared under field conditions during the regular rapeseed growing season in 2019 at Guotan village (30˚18′ N, 115˚60′ E, Wuxue, Hubei Province, China). The basic agrochemical characteristics of the soil were as follows: pH (1:1 soil: H$_2$O (w/v)), 5.18; organic matter, 37.16 g kg$^{-1}$; total N, 1.86 g kg$^{-1}$; Olsen-P, 21.70 mg kg$^{-1}$; and hot

water-soluble B, 0.10 mg kg$^{-1}$. The application rate of B fertilizer was 15 kg borax ha$^{-1}$ in the normal-B treatment, and no B fertilizer application was applied as the B-deficiency treatment. Rapeseed plants were cultivated at a distance of 15×25 cm in a 1.2×15 m plot, each variety was planted in four lines, and each treatment included three replicates. After harvest, all the seeds were allowed to dry naturally before determining the seed yield.

## Statistical analysis

The data were analyzed using Student's $t$-test, and significance was defined as $P<0.05$ or $P<0.01$. $r$ and $p$ values of the correlation analysis were determined by Pearson correlation analysis.

## Supporting information

**S1 Fig. Phenotype of wild-type and transgenic plants to normal B and low-B conditions.** (*A*) Phenotype of 15-d-old wild-type (W10) and pQ::BnaA3.NIP5;1-GFP transgenic plants grown under normal B (100 μM) condition. Scale bar, 2 cm. (*B* and *C*) Root dry weight and primary root length of W10 and pQ::BnaA3.NIP5;1-GFP transgenic plants under normal B (100 μM) condition. Data presented are mean values with s.d. (*n* = 6). (*D*) Seedlings of 15-d-old wild-type (QY10) and Q$^{RNAi}$ transgenic plants grown under normal B (100 μM) condition. Scale bar, 2 cm. (*E* and *F*) Root dry weight and primary root length of QY10 and Q$^{RNAi}$ transgenic plants under normal B (100 μM) condition. Data presented are mean values with s.d. (*n* = 6). (*G*) Phenotype of 15-d-old wild-type (W10) and W10$^{RNAi}$ lines grown under low B (0.25 μM) condition. Scale bar, 2 cm. (*H-J*) Relative expression, root dry weight and primary root length of W10 and W10$^{RNAi}$ lines under low B (0.25 μM) condition. Data presented are mean values with s.d. (*n* = 3 in (*H*) and *n* = 6 in (*I* and *J*)).
(TIF)

**S2 Fig. Seed yield of QY10 and QRNAi transgenic plants.** (*A* and *B*) Seed yield of wild-type (QY10) and Q$^{RNAi}$ transgenic plants under low-B (LB, 0.25 mg kg$^{-1}$) (*A*) and high-B (HB, 1.0 mg kg$^{-1}$) (*B*) conditions by pot culture. Data presented are mean values of four replicates with s.d. $^{**}$ $P<0.01$ (Student's $t$-test).
(TIF)

**S3 Fig. BnaA3.NIP5;1 is a homolog of AtNIP5;1.** (*A*) Phylogenetic analysis of NIP proteins in *Arabidopsis* and *B. napus*. The phylogenetic tree was constructed using MEGA 5.10 software by Neighbor-Jointing method with 1000 bootstrapping trials. (*B*) Alignment of BnaNIP5;1s with AtNIP5;1. NPA motifs and ar/R are highlighted in red.
(TIF)

**S4 Fig. Tissue and subcellular localization of BnaA3.NIP5;1 in Arabidopsis.** (*A*) Image of pQ::BnaA3.NIP5;1-GFP in *Arabidopsis* root performed by fluorescence microscopy (OLYMPUS SZX 16). Scar bar, 100 μm. (*B* and *C*) Confocal images of pQ::BnaA3.NIP5;1-GFP in *Arabidopsis* root performed by confocal microscopy (TCS SP8, Leica), FM4-64 was used as an membrane-selective tracer. Scar bar, 50 μm in (*B*) and 20 μm in (*C*).
(TIF)

**S5 Fig. $^{11}$B concentration and primary root length of QY10, W10 and NIL$^{Q}$-W10.** (*A* and *B*) $^{11}$B concentration in shoot (*A*) and root (*B*) in QY10, W10 and NIL$^{Q}$-W10. Seedlings of QY10, W10 and NIL$^{Q}$-W10 were pre-cultured with 25 μM $^{11}$B for 15 d, and then exposed to a solution containing 0 B for 1 d. Subsequently, the plants were exposed to 10 μM $^{10}$B for 1 h. Shoot and root were sampled separately and subjected to $^{10}$B and $^{11}$B determination by

ICP-MS. Data presented are mean values of with s.d. ($n$ = 3). (*C*) Primary root length of QY10, W10 and NIL$^Q$-W10 grown on 100 μM B solid medium for 5 d. Data presented are mean values of with s.d. ($n$ = 16).
(TIF)

**S6 Fig. Alignment of *BnaA3.NIP5;1$^Q$* (-1 to -2927) and *BnaA3.NIP5;1$^W$* (-1 to -3175) promoter sequence.** Red boxes indicate different TEs in *BnaA3.NIP5;1$^Q$* promoter and *BnaA3. NIP5;1$^W$* promoter. Green line indicates target repeat sequence and blue line indicates invert terminal sequence.
(DOCX)

**S7 Fig. Localization of GUS activity in pQ(2927)::GUS and pQ(3175)::GUS *Arabidopsis* transgenic plants.** Seedlings of transgenic plants were grown on solid medium containing 0.3 μM B for 10 d. GUS activity are indicated by red triangle. Scar bar, 5 mm in the whole plant image and 100 μm in root tip image.
(TIF)

**S8 Fig. Re-sequencing analysis of represent varieties used for B deficiency tolerance evaluation assay.** All varieties can be divided into two haplotype *BnaA3.NIP5;1$^Q$* and *BnaA3. NIP5;1$^W$* based on one or two CTTTC copies. Red bar indicated CTTTC repeats.
(TIF)

**S9 Fig. Primary root length and seed yield of representative varieties under low-B conditions.** (*A*) Root growth appearance of the representative varieties under low B (0.1 μM) condition for 10 d. Seedlings of representative varieties were grown on the solid medium containing 0.1 B for 5 d. Scale bar, 1 cm. (*B*) Phenotype of the representative varieties grown at B-deficient field plot. Scale bar, 20 cm.
(TIF)

**S10 Fig. Analysis of 1000 bp promoter of *AtNIP5;1*, *BnaA3.NIP5;1$^Q$* and *BnaA3.NIP5;1$^w$*.** Green box indicate the CTTTC repeats and Red boxes indicate the uORF in the 5'UTR.
(DOCX)

**S11 Fig. Calibration curves used for LA-ICP-MS analysis of B concentration.** Rapeseed seedlings were grown on 0 B solid medium for 10 d, then collected and ground into powder after dried. 0, 20, 40, 100, 200, 400, 1000 μl 10 mg/L $^{10}$B were added, respectively, and mixed with the prepared powder, after absorption for 48 h, re-dried and ground to powder. After digestion, $^{10}$B concentration was determined as described above. And then 50 mg powder was pressed under 8 atm to make standard reference material. The standard reference material was put into the laser ablation chamber and scanned together with the root tips.
(TIF)

**S1 Table. List of the primers used in this stud.**
(DOCX)

## Acknowledgments

We thank Professor Junpei Takano (Osaka Prefecture University, Japan) for kindly providing the *Arabidopsis* mutant *nip5;1–1*.

## Author Contributions

**Conceptualization:** Mingliang He, Fangsen Xu.

**Data curation:** Mingliang He, Liu Liu, Jinyao Zhang, Shou Qiu.

**Formal analysis:** Mingliang He, Cheng Zhang.

**Funding acquisition:** Shaowu Xue, Fangsen Xu.

**Investigation:** Mingliang He.

**Methodology:** Mingliang He, Fangsen Xu.

**Project administration:** Fangsen Xu.

**Resources:** Hong Wang, Guangsheng Yang, Shaowu Xue, Fangsen Xu.

**Software:** Mingliang He, Cheng Zhang.

**Supervision:** Fangsen Xu.

**Validation:** Fangsen Xu.

**Visualization:** Mingliang He, Cheng Zhang.

**Writing – original draft:** Mingliang He, Fangsen Xu.

**Writing – review & editing:** Mingliang He, Sheliang Wang, Lei Shi, Fangsen Xu.

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
