## [Decision Letter · Decision Letter 0]

2 May 2021

Dear Dr Xu,

Thank you very much for submitting your Research Article entitled 'Genetic variation of BnaA3.NIP5;1 expressing in the lateral root cap contributes to boron deficiency tolerance in Brassica napus' to PLOS Genetics.

The manuscript was fully evaluated at the editorial level and by four independent peer reviewers. The reviewers appreciated the attention to an important topic but identified some concerns that we ask you address in a revised manuscript

We therefore ask you to modify the manuscript according to the review recommendations. Your revisions should address the specific points made by each reviewer.

[LINK]

Yours sincerely,

Zhixi Tian, Ph.D

Associate Editor

PLOS Genetics

Li-Jia Qu

Section Editor: Plant Genetics

PLOS Genetics

Reviewer's Responses to Questions

**Comments to the Authors:**

Reviewer #1: In the manuscript entitled “Genetic variation of BnaA3.NIP5;1 expressing in the lateral root cap contributes to boron deficiency tolerance in Brassica napus”, the authors investigated the natural genetic variation in BnaA3.NIP5; 1 gene, which they identified but has not functionally characterized as a key determinant of low-B tolerance in B. napus in their previous studies. It was found that that a CTTTC tandem repeat in the 5’UTR of BnaA3.NIP5;1 altered its expression level, and associated with plant growth and seed yield. Though the orthologue of this gene in Arabidopsis has been well characterized, novel data and strong evidence were provided in this study to tell a novel story on the contribution of a superior allele to low-B tolerance of B. napus that requires high B supplies. Overall, the results are clear and the manuscript is easy to follow. Here I list my concerns for suggestion.

1 The CTTTC tandem repeat in the 5’UTR underlies the difference in expression levels in parental lines, NILs and natural populations. Then how does the CTTTC tandem repeat works? Does this repeat hinder the gene translation? Typically, the sequence variation in UTRs play roles on mRNA stability and translation, but not on transcription. More data should be provided to make the story clear.

2 Besides the phenotypic data on primary root length and productivity, the data on B uptake and/or accumulation are necessary to solidify the allelic difference in natural population (line 215-235) .

3 some typos:

L58 Under low-B conditions, B can be taken up plants via two different mechanisms.

L59 “Channel-mediated facilitated transport via NOD26-LIKE MAJOR INTRINSIC PROTEIN5;1 (NIP5;1), which encodes a channel protein belonging to the aquaporin family; this protein has been shown to facilitate B uptake from the soil in Arabidopsis roots.

Reviewer #2: Boron is essential for plant development and reproduction. In this study, the authors characterized the molecular mechanism of BnaA3.NIP5;1 in boron deficiency response. They found that higher expression of BnaA3.NIP5;1 in B. napus varieties conferred low-boron tolerance. Further analysis revealed that a CTTTC tandem repeat in the 5’UTR of BnaA3.NIP5;1 determined its expression level. They also convincingly demonstrated that BnaA3.NIP5;1 transport boron into root tips and promote root growth under B-deficiency condition. Overall, this study provides new insights into the low-B tolerance and a novel perspective on the genetic improvement of boron efficiency in B. napus.

However, there are still a few issues need to be addressed.

1. The authors claimed that B taken up by BnaA3.NIP5;1 was used only for local root growth, however changing the expression of BnaA3.NIP5;1 dramatically influence the shoot growth and yield. Therefore, the tissue specific expression of BnaA3.NIP5;1 should be present. Is there any possibility that the BnaA3.NIP5;1 also has function for boron transport at SAM cells, which contribute to the shoot growth. If BnaA3.NIP5;1 only expressed in lateral root cap and affected root tip boron uptake and distribution, the authors should propose a possible mechanism to explain the impact of BnaA3.NIP5;1 for shoot growth.

2. In fig3E, it's hard to distinguish the QC from other regions in root tips, it seems that high content of boron was accumulated in root meristem region.

3. Given the presence of other boron related transporters in plant, how could the higher expression of BnaA3.NIP5;1 in lateral root cap to maintain higher boron distribution in root tip under boron deficient condition and promote root growth? Possible explanation should be discussed.

Reviewer #3: Boron (B) is an essential element for plant growth and development. B nutrition is of great important for productivity of crops like rapeseed. In this manuscript He and her/his coworkers characterized a previously identified B transporter gene NIP5;1 for B-dependent root growth. They further discovered a CTTTC tandem repeat in the 5’-UTR of NIP5;1 that was capable to regulate the gene expression levels, and the corresponding natural variations were associated with root growth and seed yield under limited B conditions. This story hightlights a novel function of B transporters in root development, and provides the favorite gene alleles for breeding low-B tolerant crops. The manuscript is well written with a proper language. Some minor points are needed to be considered for improving the quality of manuscript.

Minor points:

1) Line 161-162: this sentence can be “ By contrast, the shoots development did not differ among all genotypes”.

2) Line 172-173: this sentence can be moved to the materials and methods part.

3) Line 285-285: As shown in Figure 4, the activities of pQ and pW differed when they were expressed in Arabidopsis. Thus, this type of regulation may also exist in Arabidopsis even through the absence of CTTTC element in AtNIP5;1.

4) Line 287-291: few words on the further research can be added.

Reviewer #4: Boron (B) is an essential micronutrient. But different plant species have different demand for B, among which allotetraploid rapeseed has a special high B demand. Defficiency in B supply usually caused serious yield loss in rapeseed. However, there is also a very large genetice difference in rapeseed resistance to B deficiecny, which lies the base of the current study by He et al. This is a followed study from the same lab. In their previous studies, they identified a major QTL qBED-A3a and the possible candidate gene BnaA3.NIP5;1, whose expression is higher in B effecient varity QY10, but there is no difference in amion acid sequence with that in the B inefficient varity W10. In this study, the authors tried to elucidate why BnaA3.NIP5;1 expression in QY10 is higher, which also contributes to its better growth in B deficient soil. They found that BnaA3.NIP5;1 is polarly localized in the distal plasma membrane of LRC cells to facilitate B uptake into root tips, which also clarify why QC region has a higher B concentration. The interesting finding is that A CTTTC copy deletion within the 5’UTR increased BnaA3.NIP5;1 expression levels in QY10, whereas addition of the CTTTC into QY10 decreased the expression, so this tandem repeat is closely related with plant growth and seed yield. Finally, the authors presented solid evidence in the field test with natural populations and near-isogenic lines that the varities carrying BnaA3.NIP5;1Q allete produced higher seed yield under low-B condition. Overall speaking, this is a excellent work to provide novel insight into low-B tolerance in rapeseed, and more important is that the elite allele of BnaA3.NIP5;1 could be used as a direct target for breeding low-B tolerant cultivars in future practice. All the experiments were well designed and the contents were presented logically and easy to follow.

minor suggestions:

1. Line 157-158: it is better to rephase the sentence into : we generated transgenic plants expresseing pQ::BnaA3.NIP5;1-GFP or pW::BnaA3.NIP5;1-GFP in the background of Arabidospsis mutant nip5;1-1.

2. It is interesting that B taken up by LRC sells is used only for local root growth. Is there any effect of these transformation on the seed production of the transgenic arabidospsis plants?

3. In the discussion, is it possible to give proposes how the tandem repeat of CTTTC functioning in regulating the expression of BnaA3.NIP5;1?

**Have all data underlying the figures and results presented in the manuscript been provided?**

Reviewer #1: Yes

Reviewer #2: Yes

Reviewer #3: Yes

Reviewer #4: Yes

PLOS authors have the option to publish the peer review history of their article (what does this mean?). If published, this will include your full peer review and any attached files.

Reviewer #1: No

Reviewer #2: No

Reviewer #3: No

Reviewer #4: No

---

## [Decision Letter · Decision Letter 1]

10 Jun 2021

Dear Dr Xu,

We are pleased to inform you that your manuscript entitled "Genetic variation of BnaA3.NIP5;1 expressing in the lateral root cap contributes to boron deficiency tolerance in Brassica napus" has been editorially accepted for publication in PLOS Genetics. Congratulations!

Yours sincerely,

Zhixi Tian, Ph.D

Associate Editor

PLOS Genetics

Li-Jia Qu

Section Editor: Plant Genetics

PLOS Genetics

Comments from the reviewers (if applicable):

Reviewer's Responses to Questions

**Comments to the Authors:**

Reviewer #1: All of my concerns have been well adressed. it is a nice story and I recommend this manuscript for publication.

Reviewer #2: I'm happy with the current version.

Reviewer #3: The revised version had addressed all the question I have rasied.

Reviewer #4: I am satisfactory with the revisions and have no further comments

**Have all data underlying the figures and results presented in the manuscript been provided?**

Reviewer #1: Yes

Reviewer #2: None

Reviewer #3: None

Reviewer #4: Yes

PLOS authors have the option to publish the peer review history of their article (what does this mean?). If published, this will include your full peer review and any attached files.

Reviewer #1: No

Reviewer #2: No

Reviewer #3: No

Reviewer #4: No

**Data Deposition**

http://datadryad.org/submit?journalID=pgenetics&manu=PGENETICS-D-21-00377R1

**Press Queries**

---

## [Editor Report · Acceptance letter]

28 Jun 2021

PGENETICS-D-21-00377R1 

Genetic variation of BnaA3.NIP5;1 expressing in the lateral root cap contributes to boron deficiency tolerance in Brassica napus 

Dear Dr Xu, 

We are pleased to inform you that your manuscript entitled "Genetic variation of BnaA3.NIP5;1 expressing in the lateral root cap contributes to boron deficiency tolerance in Brassica napus" has been formally accepted for publication in PLOS Genetics! Your manuscript is now with our production department and you will be notified of the publication date in due course.

With kind regards,

Andrea Szabo

PLOS Genetics

On behalf of:
